# Effect of Spray-Drying and Freeze-Drying on the Composition, Physical Properties, and Sensory Quality of Pea Processing Water (*Liluva*)

**DOI:** 10.3390/foods10061401

**Published:** 2021-06-17

**Authors:** Weijun Chen, Hoi Tung Chiu, Ziqian Feng, Evelyne Maes, Luca Serventi

**Affiliations:** 1Department of Wine, Food and Molecular Biosciences, Faculty of Agriculture and Life Sciences, Lincoln University, RFH Building, Lincoln P.O. Box 85054, Christchurch 7647, New Zealand; Weijun.Chen@lincolnuni.ac.nz (W.C.); Tyrande.Feng@lincolnuni.ac.nz (Z.F.); 2Dry Food NZ Ltd., 12 Ngati Kuia Drive, Port, Havelock 7100, New Zealand; monika@dfnz.co; 3Proteins & Metabolites Team, AgResearch Limited, Lincoln 7674, New Zealand; Evelyne.maes@agresearch.co.nz

**Keywords:** split yellow peas, soaking water, cooking water, spray-drying, freeze-drying, proximate composition, protein profile, particle size, colour, sensory

## Abstract

Spray-drying and freeze-drying can extend the shelf life and improve the transportability of high-nutritional foods such as *Liluva* (processing water of legumes). Nonetheless, the effects of these processes on nutrition, physiochemical properties, and sensory quality are unknown. In this study, particle sizes, protein profiles, colour, and preliminary sensory profile of pea powder samples were determined by Mastersizer 3000, protein gels, chroma meter, and 9-point hedonic scale, respectively. Results indicated that no significant difference was found in the molecular weight distribution of protein bands in pea water and sensory profile after drying. Fibre content in pea water after spray-drying was higher while soluble carbohydrates and minerals were lower than those after freeze-drying. Spray-drying decreased pea water’s lysine content, particle size, redness colour, and yellowness colour, while it increased its light colour; however, freeze-drying showed the opposite results. Overall, spray-drying could be a better drying technology that can be applied to dry pea water. Further experiments are required, however, to determine the influence of drying technologies on emulsifying activity.

## 1. Introduction

Legumes, such as peas, chickpeas, and beans, are low price and high nutritional value foods that are widely consumed by people all around the world. Among legumes, peas are a good source of plant-based protein because the protein content in peas is high at about 22.3 g of protein/100 g [1]. Although the antinutrients (including phytic acid, tannins, and proteolytic inhibitors) found in peas decrease the digestibility of this protein, the soaking, cooking, or baking of peas can reduce the antinutrients and improve the protein bioactivity [1].

In industrial production of legumes, wastewater is generated. *Liluva*, namely the water generated by the soaking, cooking, or canning process of legumes, can be upcycled into functional food ingredients [2]. The wastewater from soaking and cooking of 100 g split yellow peas contained about 1.89 g and 4.4 g solids respectively, including protein, soluble and insoluble carbohydrates, and minerals. In peas’ soaking and cooking water, soluble carbohydrates represented 37% and 30% of the dry matter [3,4]. Additionally, the protein content in peas’ soaking and cooking water comprises a high percentage of the dry matter as well at around 30% [3,4]. With such high contents of soluble carbohydrates and proteins, pea water has the potential to be a foaming or emulsifier agent [2], which are two important agents in baking products to increase the volume and stabilize the starch–lipid networks of bakery foods [5]. Therefore, *Liluva* is a valuable ingredient that can be applied in the food industry to increase bioactive content and improve the functional properties of food products.

In order to store and transfer pea water more conveniently, and to increase the range of applications, drying methods, such as spray-drying and freeze-drying, can be applied to remove the water and convert the pea water into a powder. Spray-drying is a thermal method widely used in the food industry to produce a dry powder from a liquid [6]. On the other hand, freeze-drying is a nonthermal method that is commonly used for the dehydration of heat-sensitive food through combining freeze and vacuum drying [7]. Even though the shelf life of peas’ soaking and cooking water may be extended by drying because of the low water activity [8], its composition and functional properties might change with drying.

The aim of this study was to analyse the effects of spray-drying and freeze-drying on peas’ soaking and cooking water, with particular emphasis on spray-drying. Proximate composition, content of free amino acids, protein bands, particle sizes, and colour were investigated in this article. Additionally, preliminary sensory analysis of a food product (sponge cake) containing peas’ raw water or reconstituted spray-dried powder was covered.

## 2. Materials and Methods

### 2.1. Sample Preparation

Split yellow peas (Cates, New Zealand), wheat flour (Pams, New Zealand), apple vinegar (DYC, New Zealand), icing sugar (Pams, New Zealand), and baking powder (Pams, New Zealand) were used in this study. The soaking water and cooking water of split yellow peas were prepared as described by Serventi [2]. Briefly, split yellow pea soaking water was prepared by soaking split yellow peas in water at a ratio of 1:3.3 (pea to water) for 17 h. After soaking, the soaked split yellow peas were cooked in water at a ratio of 1:1.75 (pea to water) for 90 min. Water samples after soaking and cooking were collected separately. Next, the soaking and cooking water of split yellow peas was spray-dried by Dry Food NZ Ltd. (Havelock, New Zealand) (the processing parameters of spray-drying technique are shown in Appendix A) and freeze-dried (Lincoln University, New Zealand).

### 2.2. Proximate Composition

The proximate composition of concentrated powders was quantified with the following methods: moisture content (AACC method 44-15A) [9], soluble carbohydrates (Pollock and Jones, Jermyn) [10,11], protein content by total nitrogen (AOAC method 954.01) [12] with a conversion factor of 6.25, fat (AOAC method 920.39) [12], ash (AOAC method 930.05) [13], and insoluble carbohydrates by difference.

### 2.3. Free Amino Acid Profile

The free amino acid profile was analysed by the Agilent 1100 series HPLC system (Agilent Technologies, Walbronn, Germany) with a 150 mm × 4.6 mm, 3 µm C-18 column (Winlab, Scotland) at 40 °C according to Heems, Luck, Fraudeau, and Verette (1998) and Carducci et al. (1996) [14,15]. The precolumn derivatization was performed on the autosampler. O-phthaldialdehyde (OPA) and 9-fluorenylmethyl chloroformate (FMOC) were used as primary and secondary amino acid derivatization reagents, respectively. The detection was performed using a fluorescence detector with the following settings: 335 nm (excitation) and 440 nm (emission). The detector was switched to second channel at 21 min to detect secondary amino acid proline, and the parameters changed to 260 nm (excitation) and 315 nm (emission). To make solvent A, 0.01 M Na_2_HPO_4_ was added with 0.8% THF and adjusted to pH = 7.5 with H_3_PO_4_, while solvent B comprised 50% methanol and 50% acetonitrile. Solvents A and B were used for the separation with the following pump gradients: 0 min, 0% B; 14 min, 40% B; 20 min, 50% B; 24 min, 100% B; 29 min, 100% B; 30 min, 0% B; 36 min, 0% B, with a flow rate of 0.7 mL/min. Sample injection volume was 12 µL.

### 2.4. Protein Analysis via SDS-PAGE

A sodium dodecyl sulfate polyacrylamide gel electrophoresis (SDS-PAGE) was performed to determine the molecular weight distribution of the protein present in the *Liluva* samples (6 samples in total, including the raw soaking and cooking water, spray-dried soaking and cooking water, and freeze-dried soaking and cooking water of split yellow peas) as described by Buhl et al. (2019) with modifications [16]. Invitrogen™ NuPAGE™ 4–12% Bis-Tris precast gels (Bio-Rad, Richmond, CA, USA) were used to evaluate the protein profile in this experiment. A molecular weight marker (10–250 kDa) was applied to estimate the molecular weight of the protein bands. Prior to heat treatment (100 °C for 5 min), 6 samples (concentration of 0.1%) were mixed 3:1 with the NuPAGE™ LDS Sample Buffer (4X) and Sample Reducing Agent. After heating the mixed solution, 8 μL of the molecular markers and 20 μL of the samples were loaded into the gel. Then, 200 V of a constant current and 45 min of running time were set for the electrophoresis, which was conducted in the running buffer (0.25 M Tris, 0.192 M glycine, 0.1% SDS). Next, the gel was stained with Commassie blue G-250 staining solution for 1 h. Afterwards, the gel was destained with the destain solution (20% Methanol, 10% Acetic acid) overnight.

### 2.5. Particle Size

The particle size of split yellow pea water and powder samples was measured with a Mastersizer 3000 (Malvern Panalytical) with constant refractive index (1.538) and absorption index (0.01). Modified from the method mentioned by Govoreanu, Saveyn, Van der Meeren, Nopens, and Vanrolleghem (2009) [17], the machines were initially rinsed with pure RO (reversed osmosis) water to stabilize size distribution. Then, samples were diluted into about 400 mL RO water to reach the acceptable obscuration limits (10–20%) in an automated flexible volume wet sample dispersion. After reaching the obscuration limits, samples’ particle size distribution was automatically measured by the machine in quintuples.

### 2.6. Colour

Modified from the method described by Yagiz, Balaban, Kristinsson, Welt, and Marshall (2009) [18], a handheld Konica Minolta CR-400 chroma meter was used to measure the colour of four *Liluva* powder samples (spray-/freeze-dried powder of pea soaking water and spray-/freeze-dried powder of pea cooking water) after calibration with a CR-A43 calibration plate. First, the powder samples were homogenized and weighed (3 g per sample) before they were poured into transparent containers. Then, the colour parameters of *Liluva* samples, including L* (lightness), a* (greenness to redness), and b* (blueness to yellowness), were measured. The surfaces of samples were touched by the light tube directly. Samples were homogenized manually after each detection, and the colour of *Liluva* powder samples were measured in triplicates.

### 2.7. Cake Preparation

Sponge cakes were made based on the recipe of Mustafa and collaborators [19]. Briefly, 110 mL of split yellow pea cooking water or spray-dried split yellow pea cooking water was mixed with 3 g of apple cider vinegar for 7 min by using a Brabantia BBEK1092 stand mixer. The mixer was started at low speed, and the maximum speed was set when the solution became foamy. After 7 min, 130 g of icing sugar was added to the mixer and mixed for 3 min at the maximum speed. Afterwards, the blend of 130 g of plant flour and 7 g of baking powder was manually mixed with the creamy foam. During mixing, flour was added into the foam three times. After adequately mixing, 110 g of each batter was weighted and poured into the baking pan and was baked at 180 °C for 15 min in a preheated Turbofan oven (Moffat Ltd., model E32 M, Rolleston, IN, USA). The baked cakes were cooled to room temperature and cut into small square-like pieces before the sensory test.

### 2.8. Sensory Analysis

Modified from the method inferred by Sveinsdottir and collaborators [20], 20 untrained participants from Lincoln University were involved in the sensory test of sponge cake. Participants were asked to taste two sponge samples (one contained raw split yellow pea cooking water; the other contained reconstituted spray-dried split yellow pea cooking water) in the sensory room (with individual booths) at Lincoln University, New Zealand. Freeze-dried samples were not considered since they were not food grade. Soaking water samples were not tested since they contained less protein than cooking water, thus making them less suitable for egg replacement. The small, square-like sponge cake samples were put into small plastic containers (without lids). Samples were labelled with digital codes that were in random order. Water and crackers were also provided for participants to clean their mouths to avoid product carry-over. All the participants were asked to evaluate the appearance, aroma, texture, taste, and overall preference of the sponge cake samples by using a 9-point hedonic scale (1—dislike extremely; 5—neither dislike nor like; 9—like extremely).

### 2.9. Data Analysis

All data was calculated and presented as average ± standard deviation by using Excel, Microsoft 365. Statistical analysis was performed by Minitab version 19. One-way analysis of variance (ANOVA) was applied to the analysis of colour, foaming ability, and emulsifying activity. Analysis of the preliminary sensory test was conducted by ANOVA using the general linear model (GLM) procedure and a post-hoc Tukey’s honest significant difference (HSD) test (*p* < 0.05).

## 3. Results and Discussion

### 3.1. Proximate Composition

According to Table 1, spray-dried split yellow pea cooking water had higher dry matter, protein, and fibre contents, but lower amounts of soluble carbohydrates and minerals than spray-dried split yellow pea soaking water. Previous studies about the compositions of freeze-dried split yellow pea cooking and soaking water presented slightly different profiles, where freeze-dried pea soaking water showed higher levels of protein (31.7%), soluble carbohydrates (36.5%), and minerals (13.8%) but lower fibre content (18%) than freeze-dried pea cooking water (28.2% of protein, 25.2% of soluble carbohydrates, 9.1% of minerals, and 34.7% of fibre) [5,21]. When comparing different drying methods under the same type of *Liluva*, spray-dried water demonstrated higher fibre but lower soluble carbohydrate and mineral levels than freeze-dried water [5,21]. In terms of the protein content, spray-dried soaking water presented a lower amount of protein than freeze-dried soaking water, while spray-dried cooking water conversely showed higher protein content than freeze-dried cooking water [5,21].

In pea soaking and cooking raw water, the content of proteins composes about 30% of the dry matter [2]. Similar protein concentrations were observed in pea powders, with 25.2% in spray-dried soaking powder, 34.6% in spray-dried cooking powder, 31.7% in freeze-dried soaking powder, and 28.2% in freeze-dried cooking powder. This is possibly because freeze-drying does not involve heating and therefore does not denature the protein. While spray-drying applies heat to the peas’ protein, the treatment time might be too short to denature proteins [22]. In terms of the higher protein content in spray-dried pea cooking water (34.6%), it is possibly because the low soluble fibre content in spray-dried pea cooking powder increased its percentage of protein content.

Additionally, as reported by Serventi [2], the content of soluble carbohydrates in split yellow pea raw soaking water was 0.69 g/100 g. In other words, the content of soluble carbohydrates made up to 37% of the dry matter. Additionally, the dry matter in split yellow pea raw cooking water also contained a high percentage of soluble carbohydrates at around 30% [2]. Compared with the results shown in Table 1, spray-drying decreased the content of soluble carbohydrates in pea soaking and cooking water from about 37 to 25.2% and 30 to 17.4% as inferred by Shishir and Chen [23]. Sugars have low molecular weight and glass transition temperature, so the mobility of sugar molecules increases with heat treatment at above 20 °C. Therefore, the reason for the lower content of soluble carbohydrates may be that the sugar in split yellow pea water sticks to the dryer when it comes across heat; thus, the recovery yield of the material decreases [23].

### 3.2. Free Amino Acids

Twenty-one free amino acids found in pea raw water and spray-/freeze-dried powder were measured and analysed (Table 2). The content of some amino acids in pea samples exceeded the detecting limits, such as glutamic acid in all samples and aspartic acid in the spray-dried and freeze-dried soaking water powder. Additionally, some of the samples contained amino acids that exceeded the quantification limits, such as cysteine in all samples except those of pea soaking water and freeze-dried pea cooking water powder.

Nosworthy and co-authors [1] reported that the contents of methionine (around 0.22% of dry matter) and cysteine (about 0.25% of dry matter) in yellow peas were limited, while lysine was abundant at around 1.69% of dry matter. This can be linked to the high content of albumin proteins in yellow peas (14 g/100 g), which are known to have many sulphur-containing amino acids and lysine in the protein sequence, which results in high lysine content [2]. Therefore, the ratio of lysine and methionine/cysteine and in pea raw water and dried powder is important.

According to Table 2, the ratio of lysine and methionine in freeze-dried pea powder (around 7.19 µM for soaking powder and about 4.53 µM for cooking powder) was higher than that in spray-dried pea powder (around 2.47 µM for soaking powder and approximately 3.61 µM for cooking powder) and pea raw water (with about 4.07 µM for soaking water and around 3.35 f µM or cooking water). This means that freeze-drying increases the content of lysine and/or decreases methionine’s content in split yellow peas. Additionally, the ratio of lysine and methionine was almost twice as high in pea raw soaking water than in spray-dried soaking water powder at about 4.07 µM and 2.47 µM, respectively. Spray-dried pea cooking water powder had a slightly higher ratio of lysine and methionine compared to pea raw cooking water (approximately 3.61 µM and 3.35 µM, respectively). As mentioned by Brishti and collaborators [24], lysine is an amino acid that is active in the occurrence of the Maillard reaction. This might explain the lower content of lysine in spray-dried powder, as spray-drying is a thermal treatment that induces the chemical reaction of amino acid and sugars in a material.

### 3.3. Protein Molecular Weight Distribution

The protein composition of pea raw water and dried powder diluted samples were visualized by SDS-PAGE (Figure 1). In general, the protein composition (estimated by protein bands in the 1D gel) of split yellow pea raw soaking water was similar to its spray-dried or freeze-dried powder diluted water samples. Similarly to pea soaking water samples, split yellow pea raw/spray-dried/freeze-dried cooking water samples showed no significant differences in the protein bands on the 1D gel, though differences in intensity between the raw water and dried water are noticeable. This illustrates that the overall protein profile in peas was not significantly affected by drying.

A comparison of the expected molecular weight of some proteins of interest with literature was made. According to Buhl and collaborators. [16], lipoxygenase corresponded with the molecular weight of 99 kDa, and albumin was located around 10 to 12 kDa. The 7S and 11S globulin units were represented in protein bands around 15 kDa and 25 kDa, respectively [24]. In the gel, low molecular weight proteins such as albumin were more intense in split yellow pea cooking water samples than in split yellow pea soaking water samples. The reason for the higher intensity of albumin protein in pea cooking water might be the high nitrogen loss of split yellow peas, which resulted from the exposure of peas’ starchy, proteinaceous endosperms into boiling water in the cooking process [2].

Additionally, split yellow pea soaking water samples contained some large molecular weight proteins that split yellow pea cooking water samples did not have, such as lipoxygenase. The disappearance of lipoxygenase in pea cooking water samples might be because of cooking, which is a heat treatment that might denature the enzymes [2]. In terms of globulins, it is a high content protein in legume seeds and act as storage proteins [2]. Similar to the findings shown by Brishti and coworkers [24] (i.e., that mung bean proteins contained lower levels of globulins), the content of the globulins in split yellow peas was low, as the gel bands were relatively very light in colour.

### 3.4. Particle Size

Regardless of the application of drying treatments or not, the particle sizes in split yellow pea cooking water were larger than in split yellow pea soaking water (Table 3). The reason is that more insoluble carbohydrates were lost during the pea boiling process compared to the soaking process. According to Serventi [2], cellulose, hemicellulose, and pectin are the insoluble polysaccharides in peas. This explains why the particle sizes in pea cooking water were larger than that in pea soaking water.

This is also demonstrated in Figure 2, where spray-drying is shown to have significantly decreased the size of large particles to below 100 μm, with the peak particle size at about 20 μm. In addition, the majority of the particles in spray-dried pea powder diluted water were distributed from about 1 to 50 μm. Additionally, freeze-drying did not much change the size of small particles in pea soaking water. However, more large-size particles were formed in split yellow pea soaking water after freeze-drying. Similar results were found in pea cooking water (Figure 3). With the application of spray-drying, more particles in spray-dried split yellow pea cooking water were distributed in the smaller size classes compared with those in split yellow pea raw cooking water. On the other hand, more large-size particles were formed in split yellow pea cooking water after freeze-drying.

The results of spray-dried yellow pea samples are in line with the findings reported by del Rio and collaborators [25], who applied spray-drying to the protein isolates of yellow pea and decreased the proteins’ particle size. The heat treatment and atomization of spray-drying could be the reason for the breaking down of the particles. Brishti and co-authors [24] also described that freeze-drying resulted in the highest particle size of mung bean protein isolates compared to other drying methods, such as spray-drying and oven drying. This is possibly because of the aggregation of particles during the production of ice crystals in freeze-drying [24]. Joshi and collaborators (2011) [21] obtained similar results in lentil protein isolates as well.

### 3.5. Colour of Powders

In general, there were significant differences among most of the powder samples in terms of lightness, redness, and yellowness. Spray-drying lightened the colour of split yellow pea soaking and cooking water more significantly than freeze-drying. The freeze-dried powder of split yellow pea soaking or cooking water, on the other hand, exhibited redder and yellower colour than the spray-dried powder of the same water material. No colour differences were observed between freeze-dried split yellow pea soaking water powder and freeze-dried split yellow pea cooking water powder.

Among four powder samples (Table 4), spray-dried powder of split yellow pea soaking water showed the lightest colour, followed by spray-dried powder of split yellow pea cooking water, at 92.1 and 90.3, respectively. Freeze-dried powder of split yellow pea soaking and cooking water had no significant difference in lightness at 81.9 and 81.0, respectively. Freeze-dried powder of split yellow pea cooking water had the reddest and yellowest colour compared to other powder samples, with 4.31 for the red colour and 22.6 for the yellow colour.

Similar results were found by Brishti and co-authors [24], who indicated that spray-dried mung bean protein isolate powder had lighter, less red and yellow colour than its freeze-dried powder. Lentils’ protein isolates were also shown darker, redder and yellower colour after freeze-drying than after spray-drying [21]. The reason for the lighter colour of the spray-dried powder was because spray-drying broke down the particles of mung bean protein isolates; thus, more light was refracted because of their larger surface area [24]. This finding indicates that light colour is correlated to the particle size of the materials. Furthermore, the higher browning index of freeze-drying compared to spray-drying also explains the lighter colour of spray-dried samples [24].

The yellow colour of yellow peas is mainly contributed by carotenoids [26]. It is possible that freeze-drying, which is a not a heat treatment, preserves the carotenoids in split yellow peas. Degradation and isomerization may occur during heat treatment, which would reduce the yellow colour of the samples [27]. Moreover, the particles may aggregate in freeze-drying [24]. The larger particle size may be the reason for the darker, yellower, and redder colour of freeze-dried powder.

### 3.6. Sensory Quality

In the sensory analysis of sponge cakes, sponge cakes made with split yellow pea raw cooking water showed no significant difference compared to sponge cakes made with the diluted water of split yellow pea spray-dried cooking powder in all sensory attributes, including appearance, aroma, taste, texture, and overall preference (Table 5). This result indicated that spray-drying did not greatly impact the sensory profile of split yellow pea cooking water.

Our research group previously reported that pea raw cooking water contained about 30 g/100 g of protein and around 30 g/100 g of soluble fibre, while spray-dried pea cooking powder diluted water had higher protein (34.6 g/100 g) but lower soluble fibre content (17.4 g/100 g) [2]. With higher protein and lower soluble fibre content, food products might be drier because of the unbalance of water distribution in food. The aroma score reported in the current study is in disagreement with a study by Avellone and coworkers [28], who investigated the effects of the spray-drying technique on wine’s quality. They indicated that the spray-drying process caused significant reductions of aroma compounds in wines. However, in the current study, the liking scores of product aroma showed no significant difference. This suggests that spray-drying could be an ideal technique to preserve the aromatic property of *Liluva*.

## 4. Conclusions

In summary, spray-drying decreased the content of soluble fibre and lysine in split yellow pea water due to sugar loss and Maillard reaction. However, spray-drying and freeze-drying did not greatly affect the protein content or protein profile of pea water, as shown by the SDS-PAGE gel (showed similar protein bands). Compared to pea raw water, the particles in pea spray-dried powder diluted water samples were mostly related to smaller size classes, while freeze-dried powder diluted water samples were the opposite, highlighting the influence of the different drying mechanisms. Aside from particle size, colour was influenced by the drying method as spray-dried powder, due to its smaller particle sizes, refracted more light. The reduction in the red and yellow colour of spray-dried powder compared to freeze-dried powder also illustrated that heat treatment might degrade some of the pigments in split yellow peas.

In the preliminary sensory test, no significant difference was found in the sensory profiles of pea raw water and powder samples. Thus, spray-drying can be used to dry *Liluva* from peas without greatly influencing peas’ properties. Future experiments are required to determine the mineral profiles of pea dried powders and to investigate their emulsifying activity to further confirm the effects of drying methods on pea composition and sensory profiles.

## Figures and Tables

**Figure 1 foods-10-01401-f001:**
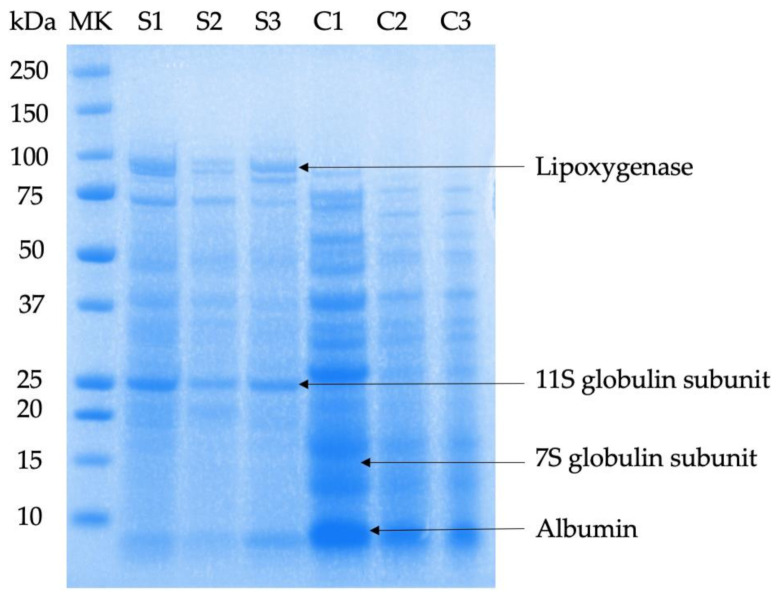
Representative SDS-PAGE gel of the samples studied. The left lane indicates molecular weight (KDa). MK—molecular markers; S1—split yellow pea raw soaking water; S2—split yellow pea spray-dried soaking water; S3—split yellow pea freeze-dried soaking water; C1—split yellow pea raw cooking water; C2—split yellow pea spray-dried cooking water; C3—split yellow pea freeze-dried cooking water.

**Figure 2 foods-10-01401-f002:**
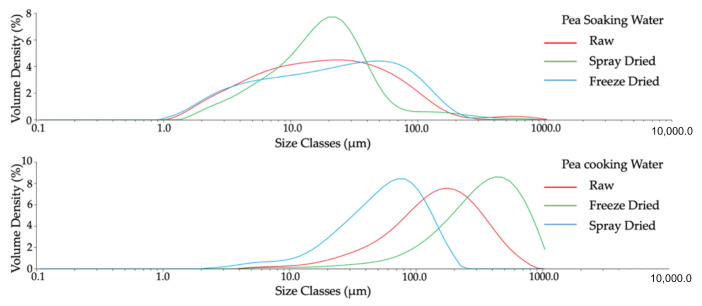
Particle size distribution of pea soaking water (**above**) and pea cooking water (**below**) in different physical states: raw, spray-dried and freeze-dried.

**Figure 3 foods-10-01401-f003:**
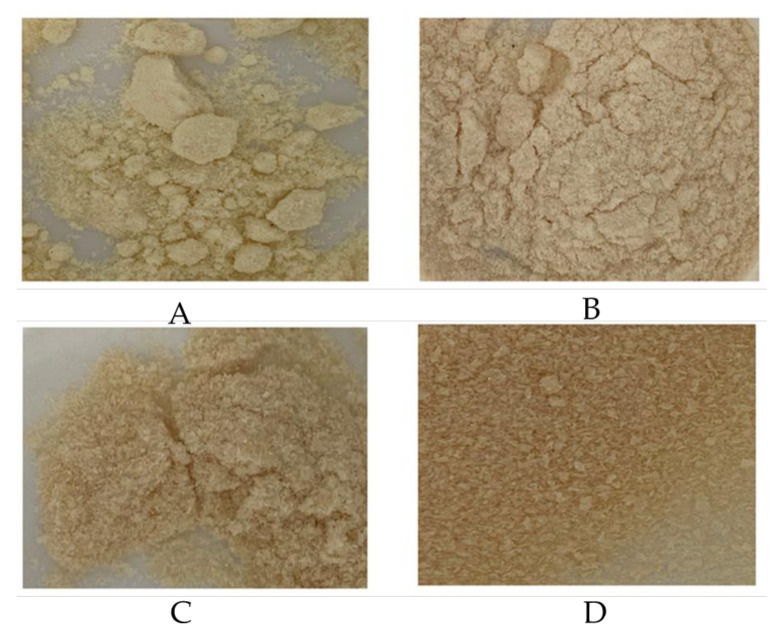
Pictures of the pea water powders. (**A**) Spray-dried pea soaking powder, (**B**) Spray-dried pea cooking powder, (**C**) Freeze-dried pea soaking powder, (**D**) Freeze-dried pea cooking powder.

**Table 1 foods-10-01401-t001:** Proximate composition of spray-dried pea soaking and cooking water powders. Different letters in the same row represent statistical difference (*p* < 0.05).

Nutrients (g/100 g)	Spray-Dried Pea Soaking Water Powder	Spray-Dried Pea Cooking Water Powder
Moisture content	7.15 ± 0.07 ^a^	4.86 ± 0.14 ^b^
Protein	25.16 ± 0.08 ^a^	34.63 ± 0.43 ^b^
Soluble carbohydrates	25.17 ± 1.70 ^a^	17.44 ± 0.92 ^b^
Insoluble carbohydrates	32.43 ± 1.82 ^a^	35.99 ± 0.98 ^b^
Minerals	10.08 ± 0.11 ^a^	7.08 ± 0.06 ^b^

**Table 2 foods-10-01401-t002:** Free amino acid profiles of different pea soaking and cooking water. Different letters represent statistical difference (*p* < 0.05).

	Soaking Water			Cooking Water		
Amino Acid	Raw (µM)	Spray-Dried (µM)	Freeze-Dried (µM)	Raw (µM)	Spray-Dried (µM)	Freeze-Dried (µM)
Asp	106.33 ± 5.21 ^b^	*	*	291.49 ± 4.33 ^a^	320.68 ± 14.42 ^a^	125.36 ± 2.96 ^b^
Glu	*	*	*	*	*	*
Cys	**	**	94.30 ± 0.74 ^b^	**	**	163.97 ± 5.09 ^a^
Asn	*	749.26 ± 16.53 ^e^	1422.57 ± 72.20 ^d^	4428.08 ± 23.66 ^a^	4144.78 ± 18.95 ^b^	2058.07 ± 14.58 ^c^
Ser	342.56 ± 8.24 ^a^	221.54 ± 0.95 ^b^	183.88 ± 7.58 ^cd^	197.30 ± 1.91 ^c^	168.69 ± 2.61 ^d^	136.06 ± 4.04 ^e^
Gln	246.78 ± 9.60 ^a^	58.57 ± 2.30 ^c^	101.47 ± 4.26 ^b^	21.71 ± 7.57 ^d^	13.97 ± 0.36 ^d^	9.48 ± 1.02 ^d^
His	128.58 ± 1.63 ^b^	192.50 ± 1.36 ^a^	86.43 ± 10.99 ^c^	127.60 ± 0.29 ^b^	121.75 ± 1.75 ^b^	100.43 ± 6.53 ^c^
Gly	*	*	558.31 ± 5.44 ^a^	288.65 ± 5.18 ^d^	379.22 ± 4.58 ^c^	420.11 ± 16.28 ^b^
Thr	376.86 ± 10.36 ^b^	542.50 ± 20.09 ^a^	267.82 ± 1.80 ^d^	325.46 ± 7.05 ^c^	261.43 ± 6.07 ^d^	238.35 ± 12.24 ^d^
Arg	*	422.27 ± 0.07	*	*	*	*
Ala	389.20 ± 5.18 ^c^	730.06 ± 0.19 ^a^	306.10 ± 1.53 ^e^	392.26 ± 0.74 ^bc^	409.71 ± 3.57 ^b^	343.02 ± 8.98 ^d^
Tau	*	*	*	609.93 ± 1.04 ^b^	643.41 ± 2.50 ^a^	*
Tyr	136.34 ± 1.79 ^a^	132.00 ± 0.08 ^a^	101.95 ± 0.86 ^c^	108.27 ± 0.51 ^b^	112.18 ± 0.03 ^b^	99.97 ± 2.66 ^c^
Val	218.58 ± 2.98 ^b^	300.00 ± 1.28 ^a^	187.71 ± 0.30 ^c^	136.62 ± 1.09 ^e^	139.43 ± 1.61 ^e^	159.75 ± 4.41 ^d^
Met	48.41 ± 0.84 ^bc^	75.07 ± 0.18 ^a^	28.75 ± 0.15 ^d^	51.31 ± 0.20 ^b^	48.89 ± 0.39 ^bc^	47.60 ± 1.62 ^c^
Try	49.39 ± 0.58 ^c^	68.18 ± 0.23 ^a^	34.65 ± 0.15 ^d^	63.31 ± 0.94 ^b^	64.81 ± 0.54 ^b^	62.10 ± 1.11 ^b^
Phe	93.89 ± 1.03 ^d^	178.84 ± 0.32 ^a^	100.00 ± 0.58 ^bc^	97.19 ± 0.04 ^cd^	103.92 ± 1.03 ^b^	98.97 ± 2.26 ^c^
Ile	97.44 ± 1.42 ^c^	142.49 ± 0.69 ^a^	101.76 ± 0.10 ^b^	64.00 ± 0.26 ^d^	66.89 ± 1.24 ^d^	63.82 ± 1.35 ^d^
Lys	197.00 ± 0.72 ^bc^	185.34 ± 1.56 ^cd^	206.75 ± 0.52 ^ab^	171.85 ± 1.65 ^e^	176.33 ± 5.96 ^de^	215.73 ± 4.32 ^a^
Leu	123.54 ± 0.94 ^b^	189.15 ± 2.18 ^a^	133.58 ± 0.98 ^b^	125.19 ± 0.32 ^b^	125.88 ± 7.53 ^b^	137.06 ± 3.41 ^b^
Pro	356.46 ± 7.78 ^a^	350.79 ± 0.84 ^a^	251.89 ± 13.09 ^b^	208.06 ± 3.01 ^bc^	232.64 ± 17.66 ^b^	173.52 ± 14.97 ^c^

* means that the amount of amino acid exceeded the detecting limits. ** means that the amount of amino acid exceeded the quantification limits.

**Table 3 foods-10-01401-t003:** The particle sizes of raw/spray-dried/freeze-dried split yellow pea soaking/cooking water.

Ingredient	Physical State	Dx (10)	Dx (50)	Dx (90)
**Pea Soaking Water**	Raw	3.59	18.1	83.0
	Spray-dried	4.91	17.8	48.6
	Freeze-dried	3.38	22.2	96.1
**Pea Cooking Water**	Raw	41.2	150.0	380.0
	Spray-dried	17.4	58.3	128.0
	Freeze-dried	103	345	737.0

**Table 4 foods-10-01401-t004:** The colour parameters (L*—lightness, a*—redness, and b*—yellowness) of spray-dried/freeze-dried split yellow pea soaking/cooking powder. Different letters represent statistical difference (*p* < 0.05).

Samples		Lightness (L*)	Redness (a*)	Yellowness (b*)
Soaking water	Spray-dried	92.1 ± 0.7 ^a^	−1.39 ± 0.05 ^d^	19.9 ± 0.3 ^c^
	Freeze-dried	81.9 ± 1.0 ^c^	2.36 ± 0.10 ^b^	20.5 ± 0.1 ^b^
Cooking water	Spray-dried	90.3 ± 0.2 ^b^	1.37 ± 0.03 ^c^	15.7 ± 0.3 ^d^
	Freeze-dried	81.0 ± 0.3 ^c^	4.31 ± 0.09 ^a^	22.6 ± 0.1 ^a^

**Table 5 foods-10-01401-t005:** Sensory profile (appearance, aroma, taste, texture, and overall preference) of sponge cakes containing split yellow pea raw cooking water (raw) or reconstituted pea cooking water powder (spray-dried). Different letters represent statistical difference (*p* < 0.05).

Recipe	Appearance	Aroma	Taste	Texture	Overall Preference
Raw	6.65 ± 0.99 ^a^	6.05 ± 1.03 ^a^	6.50 ± 1.47 ^a^	6.85 ± 1.09 ^a^	6.50 ± 1.36 ^a^
Spray-dried	6.55 ± 0.89 ^a^	6.25 ± 1.21 ^a^	6.40 ± 1.47 ^a^	6.30 ± 1.30 ^a^	6.60 ± 1.73 ^a^

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
