# Peer review of "Effect of Spray-Drying and Freeze-Drying on the Composition, Physical Properties, and Sensory Quality of Pea Processing Water (Liluva)"

_foods, 2021, doi:10.3390/foods10061401_

Round 1

Reviewer 1 Report

Spray drying and freeze drying can widely use in food processing technologies. But detailed chemical and physicochemical analysis are needed to investigate the efficiency of the different dehydration processes in details. The topic of manuscript foods-1243289 can be considered as interesting for the readers. Introduction is not complete, characteristics of spray drying and freeze drying and their effects on compounds are not given, but he main aims if the study are well defined. Methods applied in the study are adequate to the main aims of the research. Description of materials and methods are clear, but not complete. The manuscript focuses on the comparison of two drying methods, but the process parameters of spray drying and freeze drying are not provided. Manuscript contains valuable results mainly for the practice that are discussed with relevant references.

Comments, suggestions:

Please check the typos in the manuscript text (in line 14: ’ In this study, Particle sizes’, etc)

Abstract contains too general information (without comparison based on data, percentage difference etc). I suggest the rephrasing of the abstract to make it more concrete, summarizing the main essence’ of the study.

In colour measurements, had the particle size effect on the CIE LAB coordinates? Different particle size cause different reflection (surface are etc).

The parameters of spray and freeze-drying are not given. I recommend giving detailed information of the applied drying methods.

Table 1 does not contain standard deviations.

Figure 2 has low visibility. Please improve the quality of the figure.

Have the authors information about the energetic efficiency and cost of the two drying processes?

Author Response

Response to reviewer 1 comments

Point 1: Please check the typos in the manuscript text (in line 14: ’ In this study, Particle sizes’, etc)

This typo has been revised in the manuscript.

Point 2: Abstract contains too general information (without comparison based on data, percentage difference etc). I suggest the rephrasing of the abstract to make it more concrete, summarizing the main essence’ of the study.

Abstract has been supplemented with other main findings in this study, including proximate composition difference between two drying methods and difference in colour parameters between two drying methods.

Point 3: In colour measurements, had the particle size effect on the CIE LAB coordinates? Different particle size cause different reflection (surface are etc).

As explained in 3.5, the difference of particle size between spray-dried and freeze-dried samples could be the reason of the differences of lightness, redness, and yellowness indexes.

Point 4: The parameters of spray and freeze-drying are not given. I recommend giving detailed information of the applied drying methods.

The parameters of spray-drying have been added in the appendix section as this study emphasized on spray-drying.

Point 5: Table 1 does not contain standard deviations.

Standard deviations and statistical differences have been added in Table 1.

Point 6: Figure 2 has low visibility. Please improve the quality of the figure.

The visibility of Figure 2 has been improved.

Point 7: Have the authors information about the energetic efficiency and cost of the two drying processes?

The information about spray-drying has been added in the appendix section as this study emphasized on spray-drying.

Reviewer 2 Report

The manuscript entitled "Effect of spray-drying and freeze-drying on the composition, physical properties, and sensory quality of pea processing water (Liluva)" is a good work, well written in all its parts.
Spray-drying technology has become very important in recent years, as it helps the life of the food.

The authors did an excellent job comparing two relatively similar techniques: spray-drying and freeze-drying, obviously in the first technology the aromas are not significantly lost as reported in the paper "Investigation on the influence of spray-drying technology on the quality of Sicilian Nero d'Avola wines "" Food chemistry 240, pp 222-230. (add in references).
Tip for authors:
1) The authors determined the content of the minerals, it would be important to know which ones they found and if heavy metals were present, as they are fundamental parameters to be used as food, if the authors have not researched them it would be appropriate to do so, if instead it is a photo project to put it in the conclusions.
2) The authors determined the aromas through sensorial analysis, this step is very important, but it would have been more correct to go and see the aromatic fraction as in the paper "Investigation on the influence of spray-drying technology on the quality of Sicilian Nero d'Avola wines "" Food chemistry 240, pp 222-230 ", would have given added value to the manuscript.

Author Response

Response to reviewer 2 comments

Point 1: The authors determined the content of the minerals, it would be important to know which ones they found and if heavy metals were present, as they are fundamental parameters to be used as food, if the authors have not researched them, it would be appropriate to do so, if instead it is a photo project to put it in the conclusions.

In this study, only total amount of minerals have been measured, mineral profile can be analysed in further research. This has been put in the conclusion section.

Point 2: The authors determined the aromas through sensorial analysis, this step is very important, but it would have been more correct to go and see the aromatic fraction as in the paper "Investigation on the influence of spray-drying technology on the quality of Sicilian Nero d'Avola wines "" Food chemistry 240, pp 222-230 ", would have given added value to the manuscript.

Thanks for the advice! I have added the finding of this research into the manuscript to have a comparison with our results.

Round 2

Reviewer 1 Report

Manuscript foods-1243289 has an interesting topic that has relevance for the practice, as well. The manuscript is generally well written with a logic structure. Authors have revised the manuscript thoroughly according to reviewers’ comments and suggestions. Rephrasing, additional data and information added to the paper, and more detailed discussion made the manuscript clearer and more complete. After the revision the overall scientific quality of manuscript improved significantly. I accept all answers and modifications made by the authors and recommend foods-1243289 for publishing.

Reviewer 2 Report

The authors improved the manuscript with the required suggestions